# A Theory of Multimodal Learning

**Zhou Lu**
Princeton University
`zhoul@princeton.edu`

## Abstract

Human perception of the empirical world involves recognizing the diverse appearances, or 'modalities', of underlying objects. Despite the longstanding consideration of this perspective in philosophy and cognitive science, the study of multimodality remains relatively under-explored within the field of machine learning. Nevertheless, current studies of multimodal machine learning are limited to empirical practices, lacking theoretical foundations beyond heuristic arguments. An intriguing finding from the practice of multimodal learning is that a model trained on multiple modalities can outperform a finely-tuned unimodal model, even on unimodal tasks. This paper provides a theoretical framework that explains this phenomenon, by studying generalization properties of multimodal learning algorithms. We demonstrate that multimodal learning allows for a superior generalization bound compared to unimodal learning, up to a factor of $O(\sqrt{n})$, where $n$ represents the sample size. Such advantage occurs when both connection and heterogeneity exist between the modalities.

## 1 Introduction

Even before the common era, the concept of viewing an object as the collection of its appearances, has already sprouted in early philosophy. The Buddha, in the 'Diamond Sutra', separated the essence of universe from various modalities such as sight, sound, smell, taste and touch. Two centuries ago, Immanuel Kant made a further step, positing that humans perceive only the representations of 'noumena' from the empirical world. He wrote:

> *"And we indeed, rightly considering objects of sense as mere appearances, confess thereby that they are based upon a thing in itself, though we know not this thing in its internal constitution, but only know its appearances, viz., the way in which our senses are affected by this unknown something. – Prolegomena"*

From this perspective, human cognition of the world, may therefore be considered effectively equivalent to the multiple modalities of the underlying objects. The importance of multimodality extends beyond metaphysics to everyday life: children learning languages often rely on illustrations, and even mathematicians benefit from visual aids.

However, machine learning, which could be seen as the cognition of computer systems, has not fully harnessed the power of multimodality. Multimodal machine learning, which processes and learns from data with multiple modalities, remained relatively under-explored until recently. Despite the impressive success of multimodal learning in empirical applications, such as Gato [31] and GPT-4 [26], the corresponding theoretical understanding is largely absent, often limited to heuristics.

A fascinating observation from empirical multimodal learning is that a model trained with multiple modalities can outperform a finely-tuned unimodal model, even on population data of the same unimodal task. It's not immediately clear why multimodality offers such an advantage, considering that the trained model's focus is spread across different modalities.

37th Conference on Neural Information Processing Systems (NeurIPS 2023).

While it seems challenging to outperform unimodal learning asymptotically when sufficient data is available, multimodal learning can still provide an edge under a fixed data budget. Different modalities might focus on different aspects of an object, and for a specific classification problem, one modality may require a smaller sample complexity. This phenomenon often occurs with large models handling many tasks and a vast amount of training data, suggesting that:

> Training across tasks learns a common connection between modalities efficiently, allowing the model to adapt to the modality with the smallest sample complexity.

An intuitive example of how multiple modalities help is learning parametric sine functions. The samples come in the form of

$$x \in (0, 1], y = \theta x, z = \sin(1/y),$$

where $x, y$ are the two modalities and $z$ is the label. Given data from both modalities the learning problem is trivial, even with a single training data point, while learning solely on $x$ is hard albeit there is a bijective mapping between $x, y$. From a perspective of VC-dimension, there is a gap between the class of linear functions $\{\theta x\}$ and the class of parametric sine functions $\{\sin(1/\theta x)\}$, in that the former one has VC-dimension 1 while the latter one has infinite VC-dimension. More details will be provided later.

The theory problem we study in this paper, is thus how to formalize the above heuristic with provable guarantees. To this end, we examine generalization bounds of a simple multimodal ERM algorithm, which involves two parallel stages: learning a predictor $\hat{f} \in \mathcal{F}$ based on multimodal training data, and learning a connection $\hat{g} \in \mathcal{G}$ that maps one modality to another with potentially unlabeled data. During inference, the composition $\hat{f} \circ \hat{g}$ is used to perform prediction on unimodal population data.

In this setting, we prove that the learnt unimodal predictor $\hat{f} \circ \hat{g}$ can achieve vanishing generalization error against the best multimodal predictor $f^*$ as if given multiple modalities, whenever $\mathcal{G}$ is expressive enough to realize the training data. In addition, such generalization bound depends on the complexities of both hypothesis classes $\mathcal{F}, \mathcal{G}$ separately, better than unimodal approaches which typically involve the complexity of $\mathcal{F} \circ \mathcal{G}$ or a worst-case complexity of $\mathcal{F}$, up to an $O(\sqrt{n})$ factor where $n$ denotes the size of training data. On the other hand, we show a separation between multimodal and unimodal learning, by constructing a hard instance learnable by multimodal learning, in which no matter what hypothesis class is chosen for the unimodal learning problem, it's either under-expressive or over-expressive and thus incurs constant error. Putting the two pieces together, our theory suggests that with both connection and heterogeneity, multimodal learning is provably better than unimodal learning.

The paper is organized as follows. In section 2 we formalize the setting of multimodal learning and provide a motivating example. Section 3 proves a generalization upper bound of the two-stage multimodal ERM algorithm on semi-supervised multitask learning problems. The lower bound on the separation between multimodal and unimodal learning is given in section 4, then we discuss the limitations of this paper and future directions in section 5.

## 1.1 Related Works

**Theoretical Multimodal Learning**: while empirical multimodal learning has shown significant progress, theoretical studies are relatively sparse, lacking a firm foundation. Prior theoretical investigations often focus on specific settings or incorporate additional assumptions. Some of these studies adopt an information-theoretic perspective, proposing algorithms based on total correlation or utilizing partial information decomposition to quantify relationships between modalities [35, 17]. Other studies approach the problem from a multi-view setting, typically assuming that each view alone suffices for prediction [40, 1, 11, 34].

An important work in theoretical multimodal learning is [16], which also considered the advantage of multimodal learning in generalization and is the first and probably the only general theoretical result in this field so far. In particular, they considered the population risk of a representation learning based approach where learning under different subsets of modalities is performed on the same ERM objective with shared hypothesis classes. They proved that the gap between the population risks of different subsets of modalities is lower bounded by the difference between what they called the latent

representation quality, which is the best achievable population risk with the learnt representation on the chosen subset of modalities.

There are two limitations in this result: 1, there is no quantitative characterization on how large the gap between latent representation qualities can be; 2, the comparison is not-only instance-dependent, but also carried over the same hypothesis classes and doesn't exclude the possibility that the smaller subset of modalities could potentially use a different class to bypass the gap, making the lower bound somewhat restricted. We strengthen this result by showing the gap can be as large as $\Omega(1)$ (Theorem 7), even if we allow the smaller subset of modalities to use any hypothesis class. They also showed an upper bound on the excess population risk via a standard representation learning analysis, which involves the complexity of a composition of hypothesis classes $\mathcal{F} \circ \mathcal{G}$, while our analysis decouples the complexities of hypothesis classes, leading to an improved upper bound up to a factor of $O(\sqrt{n})$.

A recent work of [32] made a more fine-grained study on multimodal learning, analyzing the benefit of contrastive loss in training dynamics. They considered the aspect of optimization instead of generalization for a particular problem, focusing on the setting of a linear data-generating model. They proved that the use of contrastive loss is both sufficient and necessary for the training algorithm to learn aligned and balanced representations.

**Empirical Multimodal Learning**: the inception of multimodal learning applications dates back to the last century, initially devised to enhance speech recognition using both vision and audio [39, 25]. Multimedia is another aspect in which multimodal learning inspired new methods for indexing and searching [10, 19]. With the development of deep learning and its success in computer vision [14] and natural language processing [38], people start studying deep multimodal learning in related tasks such as generating one modality from the other [8, 15, 30]. For a more comprehensive introduction of multimodal learning, we refer the readers to the excellent survey paper [4], see also [29, 13, 18].

Recently, the power of multimodal learning was carried out in large-scale generalist models. In [31], the training data includes a wide variety of modalities such as image, text, robotics and so on. The resulting model is reported to be able to beat fine-tuned unimodal models in some tasks. In a more recent ground-breaking result, the super-large language model GPT-4 [26] makes use of not only text data available on internet, but also data from other modalities such as audio, demonstrating excellent capabilities in integrating knowledge from multiple domains.

**Representation Learning**: this field, closely related to our work, focuses on learning a common underlying representation across multiple tasks. The typical framework of representation learning involves solving an ERM problem with a composition of hypotheses $f_t \circ g$, where $f_t$ is task-specific while $g$ is the common representation. Generalization bounds of representation learning usually involve the complexity of $\mathcal{F} \circ \mathcal{G}$ or a worst-case complexity of $\mathcal{F}$.

Starting from Baxter's study [5] which gave theoretical error bounds via covering numbers on the inductive bias learning approach [36], a long line of work has followed, each improving upon and generalizing the previous results [6, 2, 21, 7, 20, 28, 27].

For more recent works we detail several representative ones here. The work of [23] studied both representation learning and transfer learning in the setting of multitask learning, achieving dimension independent generalization bounds with a chain rule on Gaussian averages [22]. For the problem of transfer learning, [37] improved the leading term of [23] by a $O(\sqrt{n})$ factor under a task diversity assumption, while [9] obtained a similar bound under low-dimension and linear function assumptions. [3] analyzed a generalized setting called contrastive learning inspired by the success of empirical language models.

## 2 Setting

In this paper, we consider a straightforward yet non-trivial case of two modalities to ensure clarity in our presentation. Formally, we denote the set of possible observations $\mathcal{S}$ to be $(\mathcal{X}, \mathcal{Y}, \mathbb{R})$. Here, each element $s \in \mathcal{S}$ constitutes a pairing of inputs from both modalities $x \in \mathcal{X} \subset \mathbb{R}^q, y \in \mathcal{Y} \subset \mathbb{R}^k$ and their associated label $z \in \mathbb{R}$, thus forming a tuple $(x, y, z)$. Assuming without loss of generality that both $\mathcal{X}$ and $\mathcal{Y}$ are contained within their respective Euclidean unit balls. Given a probability measure $\mu$ on $\mathcal{S}$ and a loss function $\ell$, the performance of a learning algorithm $\mathcal{A}$ is measured by the population loss if we interpret $\mathcal{A}$ as a function, namely

$$\mathbb{E}_{(x,y,z) \sim \mu} \ell(\mathcal{A}(x, y), z).$$

To leverage the hidden correlation between different modalities, we aim to learn both a connection function $g$ bridging the modalities and a prediction function $f$ that accepts inputs from both modalities. Although we focus on learning a connection from $\mathcal{X}$ to $\mathcal{Y}$ for simplicity, a symmetrical approach can handle the reverse direction.

In particular, we will consider learning algorithms as a composition of functions $\mathcal{A}(x, y) = f(x, g(x))$, where $f \in \mathcal{F}$ and $g \in \mathcal{G}$ represent the hypothesis classes for both functions. This form is most general and common practical forms such as fusion $\mathcal{A}(g(x), h(y))$ can be subsumed by the general form.

The goal is to identify $\hat{f}, \hat{g}$ using multi-modal training data, to minimize the excess population risk

$$\mathbb{E}_{(x,y,z)\sim\mu}\ell(\hat{f}(x, \hat{g}(x)), z) - \min_{f\in\mathcal{F}}\mathbb{E}_{(x,y,z)\sim\mu}\ell(f(x, y), z). \tag{1}$$

In this context, we compare with the optimal predictor $f^*$ as if given complete observations of both modalities, because our objective is to achieve a performance comparable to the best predictor given both modalities. The reason is that such a predictor could have a significant advantage over any unimodal predictor that does not learn these connections (either explicitly or implicitly).

We seek statistical guarantees for $\hat{f}$ and $\hat{g}$. To provide generalization bounds on the excess population risk, we require a complexity measure of hypothesis classes, defined as follows.

**Definition 1** (Gaussian average). Let $Y$ be a non-empty subset of $\mathbb{R}^n$, the Gaussian average of $Y$ is defined as

$$G(Y) = \mathbb{E}_\sigma\left[\sup_{y\in Y}\sum_{i=1}^n \sigma_i y_i\right]$$

where $\sigma_i$ are iid standard normal variables. Similarly, we can define the function version of Gaussian average. Let $\mathcal{G}$ be a function class from the domain $\mathcal{X}$ to $\mathbb{R}^k$, and $X = \{x_1, ..., x_n\}$ be the set of input sample points. We define the Gaussian average of the class $\mathcal{G}$ on sample $X$ as:

$$G(\mathcal{G}(X)) = \mathbb{E}_\sigma\left[\sup_{g\in\mathcal{G}}\sum_{i=1}^k\sum_{j=1}^n \sigma_{i,j} g_i(x_j)\right],$$

where $\sigma_{i,j}$ are iid standard normal variables.

We make the following Lipschitz assumption on the class $\mathcal{F}$ and the loss function $\ell$, which is standard in literature.

**Assumption 1.** *We assume that any function $f : \mathbb{R}^{q+k} \to \mathbb{R}$ in the class $\mathcal{F}$ is L-Lipschitz, for some constant $L > 0$. The loss function $\ell$ takes value in $[0, 1]$, and is 1-Lipschitz in the first argument for every value of $z$.*

## 2.1 A Motivating Example

To introduce our theoretical findings, we present a straightforward but insightful example, illustrating the circumstances and mechanisms through which multimodal learning can outperform unimodal learning. Despite its simplicity and informal nature, this example captures the core concept of why multimodal learning requires both connection and heterogeneity, and we believe it is as vital as the more formal statements that follow.

Consider the problem where $\mathcal{X} = \mathcal{Y} = (0, 1]$. Any potential data point $(x, y, z)$ from $\mathcal{S}$ is governed by a parameter $\theta^* \in (0, 1]$, such that

$$y = \theta^* x, z = \sin(1/y).$$

The loss function choice is flexible in this case, and any frequently used loss function like the $\ell_1$ loss will suffice.

Suppose we have prior knowledge about the structure of the problem, and we select $\mathcal{G} = g(x) = \theta x | \theta \in (0, 1]$ and $\mathcal{F} = \sin(1/x)$ as our hypothesis classes. If we have sampled data from both modalities, we can easily learn the correct hypothesis via Empirical Risk Minimization (ERM): simply take any $(x, y, z)$ sample and compute $\theta = y/x$.

However, if we only have sampled data with the $\mathcal{Y}$ modality concealed, there could be multiple $\theta$ values that minimize the empirical loss, making the learning process with $\mathcal{F} \circ \mathcal{G}$ significantly more challenging. To formalize this, we can calculate the Gaussian averages for both scenarios. In the multimodal case, the Gaussian average of $\mathcal{F}$ is zero since it's a singleton. $G(\mathcal{G}(X))$ can be upper bounded by

$$G(\mathcal{G}(X)) = \mathbb{E}_\sigma \left[ \sup_{\theta \in (0,1]} \theta \sum_{i=1}^n \sigma_i x_i \right] \leq \mathbb{E}_\sigma \left[ |\sum_{i=1}^n \sigma_i x_i| \right] = O(\sqrt{n}).$$

In contrast, the Gaussian average of $\mathcal{F} \circ \mathcal{G}$ is larger by a factor of $\sqrt{n}$

$$G(\mathcal{F} \circ \mathcal{G}(X)) = \mathbb{E}_\sigma \left[ \sup_{\theta \in (0,1]} \sum_{i=1}^n \sigma_i \sin(\frac{1}{\theta x_i}) \right] \geq \mathbb{E}_\sigma \left[ \sum_{i=1}^n \frac{1}{2} |\sigma_i| \right] = \Omega(n), \tag{2}$$

for some sample $X$ (we leave the proof in the appendix), see also [24].

This separation in Gaussian averages implies that unimodal learning can be statistically harder than multimodal learning, even if there exists a simple bijective mapping between $x, y$. We summarize the intrinsic properties of multimodal learning leading to such separation as follows:

> **Heterogeneity:** multimodal data is easier to learn than unimodal data.

> **Connection:** a mapping between multiple modalities is learnable.

Thereby, the superiority of multi-modality can be naturally decomposed into two parts: a model trained with multi-modal data performs comparably on uni-modal population data as if multi-modal data is provided (connection), a model trained and tested with multi-modal data outperforms any model given only uni-modal data (heterogeneity).

We note that both connection and heterogeneity are crucial to achieving such advantage: connection allows efficiently learning of $\mathcal{Y}$ from $\mathcal{X}$, while heterogeneity guarantees that learning with $\mathcal{X}$ is harder than learning with both $\mathcal{X}, \mathcal{Y}$. Lacking either one can lead to ill-conditioned cases: when $x \equiv y$ the connection is perfect while there is no heterogeneity and thus no need to learn anything about $\mathcal{Y}$ at all. When $x$ is a random noise the heterogeneity is large while there can't be any connection between $\mathcal{X}, \mathcal{Y}$, and it's impossible to come up with a non-trivial learner solely on $\mathcal{X}$.

**Remark 2.** The example above can be converted to be hard for each single modality. Any potential data point $(x, y, z)$ is now generated by three parameters $c \in (0, 1), \theta_1 \in (1, 2), \theta_2 \in (-2, -1)$, under the constraint that $\theta_1 + \theta_2 \neq 0$, and $(x, y, z)$ is of form $(c\theta_1, c\theta_2, c(\theta_1 + \theta_2))$. The hypothesis classes are now $\mathcal{G} = \{g(x) = \theta x, \theta \in (-1, 0) \cup (0, 1)\}$ and $\mathcal{F} = \sin(1/x)$. For any uni-modal data $x = c\theta_1$, the range of ratio $(x + y)/x$ is $(1 - 2/\theta_1, 0) \cup (0, 1 - 1/\theta_1)$. This range is a subset of $(-1, 0) \cup (0, 1)$ and we have that $\max(|1 - 2/\theta_1|, |1 - 1/\theta_1|) \geq 1/4$. As a result, $G(\mathcal{F} \circ \mathcal{G}(X))$ in this case is at least $1/4$ of that in the simpler example, thus the term remains $\Omega(n)$. On the other hand, we have that $\max(|1 - 2/\theta_1|, |1 - 1/\theta_1|) \leq 1$, so $G(\mathcal{G}(X)) = O(\sqrt{n})$ holds still. The same argument holds for $\mathcal{Y}$ similarly.

## 3 The Case of Semi-supervised Multitask Learning

The efficacy of practical multimodal learning often hinges on large models and extensive datasets, with the majority potentially being unlabeled. This is especially true when the training data encompasses a broad spectrum of tasks. Given these conditions, we are interested in the power of multimodality within the realm of semi-supervised multitask learning, where the model leverages substantial amounts of unlabeled data from various tasks.

Consider the following setup for semi-supervised multitask multimodal learning. The training data is taken from a number of tasks coming in the form of a multi-sample, partitioned into two parts $S, S'$. The labeled sample $S$ takes form $S = (S_1, ..., S_T)$ where $S_t = (s_{t1}, ..., s_{tn}) \sim \mu_t^n$, in which $\mu_t$ represents the probability measures of the $T$ different tasks from which we draw the independent data points $s_{ti} = (x_{ti}, y_{ti}, z_{ti})$ from. The unlabeled sample $S'$ takes a similar form, that $S' = (S'_1, ..., S'_T)$ where $S'_t = ((x_{t1}, y_{t1}), ..., (x_{tm}, y_{tm})) \sim \mu_{t,(x,y)}^m$, drawn independently of $S$, here by $\mu_{t,(x,y)}$ we denote the marginal distribution of $\mu_t$. We assume $S'$ has a larger size $m \gg n$ than the labeled sample $S$.

Using $S'$, we aim to learn a connection function, and with $S$, we learn a predictor that leverages both modalities. A common approach to this learning problem is empirical risk minimization (ERM). In particular, we solve the following optimization problem, where the predictors $\hat{f}_t$ on both modalities are learnt via an ERM on the labeled sample $S$,

$$\hat{f}_1, .., \hat{f}_T = \mathbf{argmin}_{f_1,...,f_T \in \mathcal{F}} \frac{1}{nT} \sum_{t=1}^{T} \sum_{i=1}^{n} \ell(f_t(x_{ti}, y_{ti}), z_{ti}).$$

Meanwhile, the connection $\hat{g}$ is learned by minimizing the distance to the true input, using the unlabeled sample $S'$ instead:

$$\hat{g} = \mathbf{argmin}_{g \in \mathcal{G}} \frac{1}{mT} \sum_{t=1}^{T} \sum_{i=1}^{m} \|g(x'_{ti}) - y'_{ti}\|.$$

Our focus is not on solving the above optimization problems (as modern deep learning techniques readily address ERM) but rather on the statistical guarantees of the solutions to these ERM problems.

To measure the performance of the solution $\hat{g}, \hat{f}_1, .., \hat{f}_T$ on the modality $\mathcal{X}$, we define the task-averaged excess risk as follows:

$$L(\hat{g}, \hat{f}_1, ..., \hat{f}_T) = \frac{1}{T} \sum_{t=1}^{T} \mathbb{E}_{(x,y,z)\sim\mu_t} \ell(\hat{f}_t(x, \hat{g}(x)), z) - \min_{f \in \mathcal{F}} \frac{1}{T} \sum_{t=1}^{T} \mathbb{E}_{(x,y,z)\sim\mu_t} \ell(f_t(x, y), z).$$

In order to bound the excess risk, it's crucial to require the class $\mathcal{G}$ to contain, at least, an approximate of a "ground truth" connection function, which maps $x$ to $y$ for any empirical observation. Later we will show that such requirement is inevitable, which can be seen a fundamental limit of our theoretical model.

**Definition 3** (Approximate realizability). We define the approximate realizability of a function class $\mathcal{G}$ on a set of input data $S = \{(x_1, y_1), ..., (x_n, y_n)\}$ as

$$\mathcal{R}(\mathcal{G}, S) = \min_{g \in \mathcal{G}} \frac{1}{n} \sum_{i=1}^{n} \|g(x_n) - y_n\|.$$

When the set $S$ is labeled, we abuse the notation $\mathcal{R}(\mathcal{G}, S)$ to denote $\mathcal{R}(\mathcal{G}, (X, Y))$ for simplicity.

We have the following theorem that bounds the generalization error of our ERM algorithm in terms of Gaussian averages and the approximate realizability.

**Theorem 4.** *For any $\delta > 0$, with probability at least $1 - \delta$ in the drawing of the samples $S, S'$, we have that*

$$L(\hat{g}, \hat{f}_1, ..., \hat{f}_T) \le \frac{\sqrt{2\pi}}{nT} \sum_{t=1}^{T} G(\mathcal{F}(\hat{X}_t, \hat{Y}_t)) + \frac{2\sqrt{2\pi}L}{mT} G(\mathcal{G}(X')) + L\mathcal{R}(\mathcal{G}, S') + (8L+4)\sqrt{\frac{\log(8/\delta)}{2nT}},$$

*where $(\hat{X}_t, \hat{Y}_t)$ denotes the set of $\{x_{ti}, \hat{g}(x_{ti})|i = 1, ..., n\}$.*

**Remark 5.** It's important to note that the Gaussian average is typically on the order of $O(\sqrt{Nd})$ when $N$ is the sample size and $d$ is the intrinsic complexity of the hypothesis class, such as the VC dimension. If we treat $d$ as a constant, for most hypothesis classes in machine learning applications, the term $G(\mathcal{G}(X'))$ typically scales as $O(\sqrt{mT})$ and each term $G(\mathcal{F}(\hat{X}_t, \hat{Y}_t))$ scales as $O(\sqrt{n})$. In practice, learning the connection $g$ is often more challenging than learning a predictor $f_t$, so it's encouraging to see the leading term $G(\mathcal{G}(X'))/mT$ decay in both $m, T$.

Theorem 4 asserts that the ERM model trained with multi-modal data, achieves low excess risk on uni-modal test data to the optimal model as if multi-modal test data is provided, when connection is learnable. This result can be naturally extended to accommodate multiple modalities in a similar way. In this case, the ERM algorithm would learn a mapping from a subset of modalities to all modalities, which involves only one hierarchy as in the two-modality case, thus our analysis naturally carries over to this new setting.

### 3.1 Necessity of A Good Connection

Recall that in the upper bound of Theorem 4, all the terms vanish as $n, T$ tend to infinity, except for $LR(\mathcal{G}, S')$. It's therefore important to determine whether the term is a defect of our analysis or a fundamental limit of our theoretical model. Here we present a simple example showing that the dependence on approximate realizability is indeed inevitable.

Let $\mathcal{X} = \mathcal{Y} = \{0, 1\}$ and $n, T \geq 2$. Each probability measure $\mu_t$ is determined by a Boolean function $b_t : \{0, 1\} \to \{0, 1\}$, and for each observation $s_t$, the label $z_t = b_t(y_t)$ is purely determined by $y_t$. In particular, the four possible observations

$$(0, 0, b_t(0)), (1, 0, b_t(0)), (0, 1, b_t(1)), (1, 1, b_t(1))$$

happen with the same probability for any $t$.

For the hypothesis classes, $\mathcal{G}$ includes all Boolean functions $g : \{0, 1\} \to \{0, 1\}$, while $\mathcal{F}$ includes all 1-Lipschitz functions $\mathbb{R}^2 \to \mathbb{R}$. It's straightforward to verify that

$$LR(\mathcal{G}, S) = \frac{c_0}{nT}\left(\frac{1}{2} - \frac{|\sum_{i=1}^{c_0} \sigma_i|}{2c_0}\right) + \frac{c_1}{nT}\left(\frac{1}{2} - \frac{|\sum_{i=1}^{c_1} \sigma_i|}{2c_1}\right),$$

where $\sigma_i$ are iid Rademacher random variables, and $c_0, c_1$ denotes the number of observations with $x = 0$ and $x = 1$ respectively. The loss function $\ell$ is set as $|f(x, y) - z|$.

The simplest version of Bernstein inequality states that for any $\epsilon > 0$ and $m \in \mathbb{N}^+$

$$\mathbb{P}\left(\frac{1}{m}|\sum_{i=1}^{m} \sigma_i| > \epsilon\right) \leq 2e^{-\frac{m\epsilon^2}{2(1+\frac{\epsilon}{3})}},$$

therefore with probability at least $\frac{3}{4}$, we have that $|c_0 - c_1| \leq 3\sqrt{nT}$ since $|c_0 - c_1|$ itself can be written in the form of $|\sum_{i=1}^{nT} \sigma_i|$.

Condition on $|c_0 - c_1| \leq 3\sqrt{nT}$, we have that $c_0, c_1 \leq \frac{nT}{2} + 2\sqrt{nT} \leq \frac{3nT}{4}$. Using the Bernstein inequality again, with probability at least $\frac{7}{8}$, it holds $|\sum_{i=1}^{c_0} \sigma_i| \leq 8\sqrt{c_0}$ and similarly for $c_1$. Putting it together, with probability at least $\frac{1}{2}$, the term $LR(\mathcal{G}, S)$ can be lower bounded as

$$LR(\mathcal{G}, S) \geq \frac{1}{2} - \frac{4\sqrt{3}}{\sqrt{nT}}.$$

On the other hand, the population loss of any $f(x, g(x))$ composition is clearly at least $\frac{1}{2}$ because the label $z$ is independent of $x$, while the population loss of $\{f_t(x, y)\}$ with the choice of $f_t(x, y) = b_t(y)$ is zero. As a result the excess risk on population is at least $\frac{1}{2}$ which doesn't scale with $n, T$. When $n, T$ are large enough, the term $LR(\mathcal{G}, S)$ matches the optimal achievable excess risk.

## 4  The Role of Heterogeneity

So far we have demonstrated that as long as a good connection indeed exists, learning with multimodal training data using a simple ERM algorithm yields a unimodal model which is guaranteed to perform as well as the best model $f_t^*(x, y)$ with both modalities. The story is not yet complete. In order to explain the empirical phenomenon we're investigating, we still need to determine in what circumstance learning with multimodal data is strictly easier than unimodal learning. A good connection itself isn't sufficient: in the case of $y \equiv x$ which admits a perfect connection function, bringing $\mathcal{Y}$ into consideration apparently gives no advantage.

To address this question, we move our eyes on heterogeneity, another fundamental property of multimodal learning that describes how modalities diverge and complement each other. Intuitively, heterogeneity can potentially lead to a separation between learnability in the following way: learning from a single modality is much harder than learning from both modalities, in the sense that it requires a much more complicated hypothesis class. As a result, either the sample complexity is huge due to a complicated hypothesis class, or the hypothesis class is so simple that even the best hypothesis performs poorly on population.

Consequently, we compare not only the best achievable population risks, but also the generalization errors. For unimodal learning denote $\mathcal{G}$ as the hypothesis class, we consider the ERM solution

$$\tilde{g} = \arg\min_{g \in G} \frac{1}{n} \sum_{i=1}^{n} \ell(g(x), z).$$

The generalization error of $\tilde{g}$ can be bounded via Gaussian average of the hypothesis class $\mathcal{G}$ in the following way if we denote $g^* = \text{argmin}_{g \in \mathcal{G}} \mathbb{E}_{(x,y,z) \sim \mu} \ell(g(x), z)$:

$$\mathbb{E}_{(x,y,z) \sim \mu} \ell(\tilde{g}(x), z) - \mathbb{E}_{(x,y,z) \sim \mu} \ell(g^*(x), z) \leq \tilde{O}\left(\frac{G(\mathcal{G}(X))}{n}\right),$$

which is tight in general without additional assumptions. For multimodal learning, $\mathcal{F}$ and $f^*$ can be defined in the same way.

We are going to show that, either the intrinsic gap of risk between unimodality and multimodality

$$\mathbb{E}_{(x,y,z) \sim \mu} \ell(g^*(x), z) - \mathbb{E}_{(x,y,z) \sim \mu} \ell(f^*(x, y), z) \tag{3}$$

is substantial, or the Gaussian average gap is large. This implies a separation between multimodal learning and unimodal learning, when heterogeneity is present. Consequently, we define the heterogeneity gap as follow.

**Definition 6** (Heterogeneity gap). Given a fixed hypothesis class $\mathcal{F}$ and number of sample $n \geq 2$, the heterogeneity gap between w.r.t some distribution $\mu$ and hypothesis class $\mathcal{G}$ is defined as

$$H(\mu, \mathcal{G}) = \left[ \mathbb{E} \frac{G(\mathcal{G}(X))}{n} + \mathbb{E}_{(x,y,z) \sim \mu} \ell(g^*(x), z) \right] - \left[ \mathbb{E} \frac{G(\mathcal{F}(X, Y))}{n} + \mathbb{E}_{(x,y,z) \sim \mu} \ell(f^*(x, y), z) \right].$$

The above definition measures the population risks between learning with a single modality $\mathcal{X}$ or with both modalities $\mathcal{X}, \mathcal{Y}$. When $H(\mu, \mathcal{G})$ is large, unimodal learning is harder since ERM is arguably the optimal algorithm in general. As long as Theorem 4 admits a low risk, the heterogeneity gap itself directly implies the superiority of multi-modality by definition. Therefore, the only question left is whether such desired instance (large heterogeneity gap + perfect connection) actually exists.

To this end, the following theorem provides the existence of such instance, proving that our theory is indeed effective. Let's sightly abuse the notation $L(\tilde{g})$ to denote the excess population risk of $\tilde{g}$ which is the output of the unimodal ERM. We have the following lower bound w.r.t heterogeneity gap.

**Theorem 7.** *There exist $\mathcal{X}, \mathcal{Y}, \mathcal{F}$ and a class $U$ of distributions on $(\mathcal{X}, \mathcal{Y}, \mathbb{R})$, such that*

$$\forall \mathcal{G}, \forall n \in \mathbb{N}^+, \exists \mu \in U, s.t. \, H(\mu, \mathcal{G}) = \Omega(1), and \, \mathbb{E}_X[L(\tilde{g})] = \Omega(1).$$

*Meanwhile, let $\hat{f}, \hat{g}$ denote the outputs of the multimodal ERM algorithm in section 3, we have that*

$$\exists \mathcal{G}, s.t. \, \forall \mu \in U, \forall n \in \mathbb{N}^+, L(\hat{g}, \hat{f}) = 0.$$

**Remark 8.** We compare with the work of [16]. They showed a similar separation in terms of the intrinsic gap (the difference between optimal hypotheses), under a representation learning framework. We make a step further by taking the Gaussian average (how hard to learn the optimal hypothesis) into consideration, which is crucial to the $\Omega(1)$ gap in the more general setting.

Theorem 7 shows the existence of hard instances, where not only the heterogeneity gap is large, but the difference between actual risks is also substantial. It implies that under certain circumstances, multimodal learning is statistically easy, while unimodal learning incurs constant error no matter what hypothesis classes are used.

To sum up, our theory demonstrates that the superiority of multi-modality can be explained as the combined impact of connection and heterogeneity: when connection (Theorem 4) and heterogeneity (Theorem 7) both exist, multimodal learning has an edge over unimodal learning even if tested on unimodal data, providing an explanation to the empirical findings. Nevertheless, our theory also suggests a simple principle potentially useful for guiding empirical multimodal learning:

1. Collect numerous unlabeled multimodal data. Learn a connection via generative models.
2. Learn a predictor based on a modest amount of labeled multimodal data.

Such framework can be easily carried out with modern deep learning algorithms, for example we can learn the connection by generative models [12, 33].

## 4.1 Comparison with Representation Learning

It's possible to learn $\hat{f}, \hat{g}$ and minimize 1, based on observations of a single modality $\mathcal{X}$, via representation learning. In particular, representation learning solves the following unimodal ERM problem by treating $y$ as an unknown representation of $x$, on a labeled sample $S$ where $y$ is hidden

$$\hat{g}, \hat{f}_1, .., \hat{f}_T = \mathbf{argmin}_{g \in \mathcal{G}, f_1, ..., f_T \in \mathcal{F}} \frac{1}{nT} \sum_{t=1}^{T} \sum_{i=1}^{n} \ell(f_t(x_{ti}, g(x_{ti})), z_{ti}).$$

Unfortunately, although this representation learning method uses the same set of hypotheses, it leads to a worse sample complexity bound. Such method fails to exploit two essential parts of semi-supervised multimodal learning, namely the rich observation of unlabeled data $S'$, and separate learning of $f, g$. Failing to utilize $S'$ will lead to a worse factor $G(\mathcal{G}(X))/nT$ which scales as $1/\sqrt{nT}$, while in Theorem 4 the factor $G(\mathcal{G}(X'))/mT$ scales as $1/\sqrt{mT}$ which is much smaller than $1/\sqrt{nT}$.

Failing to exploit the "explicit representations" $y$ from the training data requires learning a composition $f \circ g$ from scratch, which typically leads to a worst case Gaussian average term, for example in [37] they have $\max_{g \in \mathcal{G}} G(\mathcal{F}(S(g)))$ instead where $S(g) = \{(x_{ti}, g(x_{ti}))\}$ is a random set induced by $g$. As a comparison, our approach decouples the learning of $f, g$, and the Gaussian average term $G(\mathcal{F}(\hat{X}_t, \hat{Y}_t))$ is only measured over the "instance" $\hat{S}$ which can be smaller. In fact, $G(\mathcal{F}(\hat{X}_t, \hat{Y}_t))$ can be smaller than $\max_{g \in \mathcal{G}} G(\mathcal{F}(S(g)))$ up to a factor of $O(\sqrt{n})$, see the example in appendix.

## 5 Limitations and Future Directions

In this paper we propose a theoretical framework on explaining the empirical success of multimodal learning, serving as a stepping stone towards more in-depth understanding and development of the theory. Nevertheless, as a preliminary study on the relatively unexplored field of theoretical multimodal learning, our result comes with limitations, and potentially opportunities for future research as well. We elaborate on these points below.

**More natural assumptions**: the current set of assumptions, while generally consistent with practical scenarios, does not fully align with them. Although most assumptions are satisfied in practice, the assumption on $\mathcal{F}$ containing only Lipschitz functions, is restrictive: the predictor class $\mathcal{F}$ typically comprises deep neural networks. Future work could seek to overcome the Lipschitz assumption, which is not only fundamental to our results but also a cornerstone in representation learning theory.

**Hypothesis-independent definitions**: our study characterizes multimodal learning through two central properties: connection and heterogeneity. However, their current definitions, $\mathcal{R}(\mathcal{G}, S)$ and $H(\mu, \mathcal{G})$, depend on both the data and the hypothesis class $\mathcal{G}$. It's interesting to see if we can develop theories with hypothesis-independent definitions, such as mutual information or correlation. A potential difficulty is that such statistical notions are not totally aligned with the perspective of machine learning, for example it could happen that the two modalities are independent with zero mutual information, while there exists a trivial connection mapping.

**Fine-grained analysis**: our theory, which focuses on the statistical guarantees of ERM solutions, is fairly abstract and does not make specific assumptions about the learning problem. To study other more concrete multimodal learning algorithms, for example the subspace learning method, we need more fine-grained analysis which takes the particular algorithm into account.

**More realistic examples**: while the example in section 2.1 provides theoretical justification, it remains somewhat artificial and diverges from typical multimodal learning problem structures. Ideally, we would like to see examples that closely mirror real-world scenarios. For instance, can we progress beyond the current example to provide one that captures the inherent structures of NLP and CV data?

**Benefit in optimization**: our work demonstrates the advantages of multimodal learning primarily in terms of generalization properties. However, optimization, another crucial aspect of machine learning, could also benefit from multimodality. One potential direction to explore is the possibility that multimodal data is more separable and thus easier to optimize. We provide a simple example

in the appendix that demonstrates the existence of linearly separable multimodal data, where the decision boundary for any single modality is arbitrary.

## 6 Conclusion

In this paper we study multimodal learning from a perspective of generalization. By adopting classic tools in statistic learning theory on this new problem, we prove upper and lower bounds on the population risk of a simple multimodal ERM algorithm. Compared with previous works, our framework improves the upper bound up to an $O(\sqrt{n})$ factor by decoupling the learning of hypotheses, and gives a quantitative example in the separation between multimodal and unimodal learning. Our results relate the heuristic concepts connection and heterogeneity to a provable statistical guarantee, providing an explanation to an important phenomenon in empirical multimodal learning, that a multimodal model can beat a fine-tuned unimodal one. We hope our result, though being a preliminary step into a deeper understanding of multimodal learning, can shed some light on the directions of future theoretical studies.

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
