# A    Proof of Inequality 2

*Proof.* It suffices to proving that there exist some $a_1 > a_2 > ... > a_n \geq 1$, for any $\delta_i = \pm 1$ where $i = 1, ..., n$, we can find $b \in [1, \infty)$ such that for any $i$ there exists an integer $k$ satisfying

$$ba_i - 2k\pi \in \delta_i[\frac{\pi}{4}, \frac{3\pi}{4}].$$

Let $b = 2\pi c$, the problem can be reduced to finding $c \in [1, \infty)$ such that for any $i$

$$\lfloor ca_i \rfloor \in \frac{1}{2} + \delta_i[\frac{1}{8}, \frac{3}{8}].$$

Let $a_i = 1 + \frac{1}{16^i}$. For any set of $\delta_i$, we consider $c$ of the following form

$$c = \frac{1}{2} + \sum_{i=1}^{n}(1 + c_i)16^i.$$

where we set $c_i = \frac{1}{4}\delta_i$. Clearly $\lfloor c \rfloor = \frac{1}{2}$ and $c > 1$.

Now we can write the target floor function as

$$\lfloor ca_i \rfloor = \lfloor \frac{1}{2} + \frac{1}{2 \times 16^i} + c_i + \frac{1}{16^i}\sum_{j=1}^{i-1}(1 + c_j)16^j \rfloor.$$

We estimate the absolute value of each term appearing in RHS. Apparently, $|\frac{1}{2 \times 16^i}| \leq \frac{1}{32}$. We have the following estimate

$$0 \leq \frac{1}{16^i}\sum_{j=1}^{i-1}(1 + c_j)16^j \leq \frac{5}{4}\sum_{j=1}^{i-1}\frac{1}{16^j} \leq \frac{1}{12}.$$

Because $\frac{1}{2} + \frac{1}{32} + \frac{1}{4} + \frac{1}{12} < 1$, we can drop the floor function in RHS and get

$$\lfloor ca_i \rfloor = \frac{1}{2} + \frac{1}{2 \times 16^i} + c_i + \frac{1}{16^i}\sum_{j=1}^{i-1}(1 + c_j)16^j.$$

Now, for any $\delta_i = 1$, we can check that

$$\lfloor ca_i \rfloor \in \frac{3}{4} + [-\frac{1}{32} - \frac{1}{12}, \frac{1}{32} + \frac{1}{12}] \in \frac{3}{4} + [-\frac{1}{8}, \frac{1}{8}] = [\frac{5}{8}, \frac{7}{8}].$$

Similarly, for any $\delta_i = -1$, we have that

$$\lfloor ca_i \rfloor \in [\frac{1}{8}, \frac{3}{8}].$$

This proves our claim.

$\square$

# B    Proof of Theorem 4

*Proof.* We first establish the ability of $\hat{g}$ trained on $S'$, to approximate $y_{ti}$ using $\hat{g}(x_{ti})$ on $S$. Recall the classic tool in proving uniform bounds on the estimation error in terms of Gaussian averages:

**Theorem 9** ([2, 22]). *Let $\mathcal{F}$ be a function class $f : \mathcal{X} \to [0, 1]^T$, and $\mu_1, ..., \mu_T$ be probability measures on $\mathcal{X}$ with $X = (X_1, ..., X_T) \sim \Pi_{t=1}^{T}(\mu_t)^n$ where $X_t = \{x_{t1}, ..., x_{tn}\}$. Then with probability at least $1 - \delta$ in the drawing of $X$, for all $f \in \mathcal{F}$ we have that*

$$\frac{1}{T}\sum_{t=1}^{T}\mathbb{E}_{x \sim \mu_t}[f_t(x)] - \frac{1}{nT}\sum_{t=1}^{T}\sum_{i=1}^{n}f_t(x_{ti}) \leq \frac{\sqrt{2\pi}}{nT}G(\mathcal{F}(X)) + \sqrt{\frac{9\log(2/\delta)}{nT}}.$$

By the definition of $\hat{g}$, we have that

$$\frac{1}{mT}\sum_{t=1}^{T}\sum_{i=1}^{m}\|\hat{g}(x'_{ti}) - y'_{ti}\| \leq \mathcal{R}(\mathcal{G}, X')$$

It's tempting to treat $\|\hat{g}(x) - y\|/2$ as a function of input $(x, y)$ and directly apply the above theorem, but in order to get Gaussian average of $\mathcal{G}$, we need to decouple $\|\cdot\|$ and $\hat{g}$. To this end we need the following composition property of Gaussian average

**Theorem 10** ([24]). *Let $Y \subset \mathbb{R}^n$ and $h : Y \to \mathbb{R}^m$ be a L-Lipschitz mapping. Then*

$$G(h(Y)) \leq LG(Y).$$

Because the norm function $\|\cdot\|$ is 1-Lipschitz, using Theorem 10 with $Y = \{\hat{g}(x'_{ti}) - y'_{ti}|t = 1, ..., T, i = 1, ..., m\}$ and noticing $y'_{ti}$ is fixed, we have that with probability at least $1 - \delta/4$ in the drawing of $S'$,

$$\frac{1}{nT}\mathbf{E}_S\left[\sum_{t=1}^{T}\sum_{i=1}^{n}\|\hat{g}(x_{ti}) - y_{ti}\|\right] \leq \mathcal{R}(\mathcal{G}, X') + \frac{2\sqrt{2\pi}}{mT}G(\mathcal{G}(X')) + 2\sqrt{\frac{9\log(8/\delta)}{2mT}}, \quad (4)$$

since $\frac{1}{n}\mathbf{E}_S\left[\sum_{i=1}^{n}\|\hat{g}(x_{ti}) - y_{ti}\|\right] = \mathbb{E}_{(x,y)\sim\mu_t}[\|\hat{g}(x) - y\|]$.

Now condition on the event that inequality 4 holds. To get a high probability bound on the actual drawn set $S$, we use Hoeffding's inequality since the LHS involves only the sum of bounded independent random variables. To be precise, denote $v_{ti} = \|\hat{g}(x_{ti}) - y_{ti}\|/2$, we can check that they are independent random variables, bounded in $[0, 1]$. The Hoeffding's inequality states that for any $t > 0$,

$$\mathbb{P}\left(|\sum_{t,i}v_{ti} - \mathbb{E}[\sum_{t,i}v_{ti}]| \geq t\right) \leq 2e^{-\frac{2t^2}{nT}}.$$

Taking $t = \sqrt{\frac{nT\log(8/\delta)}{2}}$, we have that with probability at least $1 - \frac{\delta}{4}$ in the drawing of $S$,

$$\frac{1}{nT}\sum_{t=1}^{T}\sum_{i=1}^{n}\|\hat{g}(x_{ti}) - y_{ti}\| \leq \mathcal{R}(\mathcal{G}, X') + \frac{2\sqrt{2\pi}}{mT}G(\mathcal{G}(X')) + 8\sqrt{\frac{\log(8/\delta)}{2mT}}.$$

It ensures that with high probability, the connection $\hat{g}$ learnt on $S'$, also performs well on the other set $S$, allowing us to further bound the estimation error of $\hat{f}$. To this end, define $f_t^* = \mathbf{argmin}_{f\in\mathcal{F}}\mathbb{E}_{(x,y,z)\sim\mu_t}\ell(f_t^*(x, y), z)$. We make a standard decomposition in proving generalization bounds:

$$L(\hat{g}, \hat{f}_1, ..., \hat{f}_T) = \left(\frac{1}{T}\sum_{t=1}^{T}\mathbb{E}_{(x,y,z)\sim\mu_t}\ell(\hat{f}_t(x, \hat{g}(x)), z) - \frac{1}{nT}\sum_{t=1}^{T}\sum_{i=1}^{n}\ell(\hat{f}_t(x_{ti}, \hat{g}(x_{ti})), z_{ti})\right)$$

$$+ \left(\frac{1}{nT}\sum_{t=1}^{T}\sum_{i=1}^{n}\ell(\hat{f}_t(x_{ti}, \hat{g}(x_{ti})), z_{ti}) - \frac{1}{nT}\sum_{t=1}^{T}\sum_{i=1}^{n}\ell(\hat{f}_t(x_{ti}, y_{ti}), z_{ti})\right)$$

$$+ \left(\frac{1}{nT}\sum_{t=1}^{T}\sum_{i=1}^{n}\ell(\hat{f}_t(x_{ti}, y_{ti}), z_{ti}) - \frac{1}{nT}\sum_{t=1}^{T}\sum_{i=1}^{n}\ell(f_t^*(x_{ti}, y_{ti}), z_{ti})\right)$$

$$+ \left(\frac{1}{nT}\sum_{t=1}^{T}\sum_{i=1}^{n}\ell(f_t^*(x_{ti}, y_{ti}), z_{ti}) - \frac{1}{T}\sum_{t=1}^{T}\mathbb{E}_{(x,y,z)\sim\mu_t}\ell(f_t^*(x, y), z)\right).$$

Each term is bounded separately. From our previous analysis of $\hat{g}$, the second term is bounded by

$$L\left(\mathcal{R}(\mathcal{G}, X') + \frac{2\sqrt{2\pi}}{mT}\mathbb{E}_{S'}G(\mathcal{G}(X')) + 8\sqrt{\frac{\log(8/\delta)}{2mT}}\right)$$

with probability at least $1 - \delta/4$ in the drawing of $S$, noticing that Lipschitzness is preserved by function composition using Assumption 1. The third term is non-positive by the optimality of $\hat{f}_t$ on

training data $S$. The last term is also bounded via Hoeffding's inequality, that with probability at least $1 - \frac{\delta}{4}$ in the drawing of $S$, this term is bounded by $\sqrt{\frac{\log(8/\delta)}{2nT}}$.

As for the first term, we again apply Theorem 9 to give uniform bounds on the task-averaged estimation error. In particular, we have that with probability at least $1 - \delta/4$ in the drawing of $S$, the first term is bounded by

$$\frac{\sqrt{2\pi}}{nT} \sum_{t=1}^{T} G(\mathcal{F}(\hat{X}_t, \hat{Y}_t)) + \sqrt{\frac{9\log(8/\delta)}{2nT}},$$

where $\hat{S} = \{(x_{ti}, \hat{g}(x_{ti}), z_{ti}) | i = 1, ..., n, t = 1, ..., T\}$ is the modification of the set $S$ by replacing each $y_{ti}$ with $\hat{g}(x_{ti})$. It's important to have $\hat{g}$ independent of $S$, so that we are able to condition on a fixed $\hat{g}$ and incorporate $\hat{g}(x)$ into the distribution $\mu$, treating $\hat{f}_t$ as a function of $x$ instead of $(x, y)$. In particular, by Theorem 9 we can bound the first term by

$$\frac{\sqrt{2\pi}}{nT} G(\tilde{S}) + \sqrt{\frac{9\log(8/\delta)}{2nT}},$$

where $\tilde{S} = \{\ell(f_t(x_{ti}, \hat{g}(x_{ti})), z_{ti}) | f_1, ..., f_T \in \mathcal{F}\}$. By the Lipschitz-ness of $\ell$ and Theorem 10, we can unroll $\ell$ and bound $G(\tilde{S})$ by

$$G(\tilde{S}) \leq G(\{f_t(x_{ti}, \hat{g}(x_{ti})) | f_1, ..., f_T \in \mathcal{F}\})$$

$$\leq \sum_{t=1}^{T} G(\{f_t(x_{ti}, \hat{g}(x_{ti})) | f_t \in \mathcal{F}\})$$

$$= \sum_{t=1}^{T} G(\mathcal{F}(\hat{X}_t, \hat{Y}_t)).$$

Finally, the use of union bound yields the claimed bound. □

## C   Proof of Theorem 7

*Proof.* We consider the same example as in section 2.1, where $\mathcal{X} = \mathcal{Y} = (0, 1]$. Any potential sample $(x, y, z)$ from $\mathcal{S}$ is governed by a parameter $\theta^* \in (0, 1]$, such that

$$y = \theta^* x, z = \sin(1/y).$$

We choose the simple loss function $\ell(x, z) = |x - z|$. Since the hypothesis class $\mathcal{F}$ is a singleton, the Gaussian average $G(\mathcal{F}(X, Y)) \equiv 0$. By the definition of $z$, we have also $\ell(f^*(x, y), z) \equiv 0$, thus the heterogeneity gap $H(\mu, \mathcal{G})$ is just

$$\mathbb{E}_X \frac{G(\mathcal{G}(X))}{n} + \mathbb{E}_{(x,y,z) \sim \mu} \ell(g^*(x), z)$$

To bound this term, we discuss two cases.

**Case 1**: $G(\mathcal{G}(X)) \geq \frac{n}{4}$ for each $X$ which is a size-$n$ subset of $\{x_1, ..., x_m\}$ where $x_i = \frac{1}{1+1/16^i}$ and $m \geq n^3$. We set $\mu$ to be the induced distribution of sampling $x$ from $\{x_1, ..., x_m\}$ under a uniform distribution, with $y, z$ determined by some $\theta^*$.

Whenever $X \sim \mu^n$ contains no duplicates, we have that $G(\mathcal{G}(X)) \geq \frac{n}{4}$, and thus $H(\mu, \mathcal{G})$ is lower bounded by $\frac{1}{4}$ because the loss function is non-negative. The only step left is how to lower bound the probability of the event that $X \sim \mu^n$ contains no duplicates.

This is exactly the classical birthday problem. The probability of the event happening is

$$\prod_{i=1}^{n} \frac{n^3 - i}{n^3} \geq (\frac{n^2 - 1}{n^2})^n = (1 - \frac{1}{n^2})^n \geq 1 - \frac{n}{n^2} \geq \frac{1}{2}.$$

As a result, $H(\mu, \mathcal{G})$ is lower bounded by $\frac{1}{8}$ in this case.

To further strengthen this "lower bound on upper bound" into a "lower bound on actual risk", we recall the construction of example 2. In that construction, for any finite subset of $\{\frac{1}{1+1/16^i}|i \in \mathbb{N}^+\}$, including $\{x_1, ..., x_m\}$, there exists a $\theta^*$ that approximately shatters the set.

In addition, the construction can be modularized for each $x_i$: to change the sign $\delta_i$, we only need to change the corresponding $c_i$ which doesn't affect other $x_j, j \neq i$. This indicates no algorithm can distinguish between two $\theta_1, \theta_2$ that induce the same labeling on $X$, but may differ arbitrarily on $\{x_1, ..., x_m\} \setminus X$ which is a larger set.

As a result, denote the set of $\theta$ that shatters $\{x_1, ..., x_m\}$ as $\mathcal{T}$ with size $2^m$, then if we draw $\theta^*$ from $\mathcal{T}$ uniformly, we have that

$$\mathbb{E}_{\theta^*} \mathbb{E}_X L(\tilde{g}) \geq \frac{m-n}{2m}.$$

Therefore, there musts exist a particular $\theta^* \in \mathcal{T}$ such that $\mathbb{E}_X L(\tilde{g}) \geq \frac{m-n}{2m} \geq \frac{1}{4}$.

**Case 2**: $G(\mathcal{G}(X)) < \frac{n}{4}$ for some $X$ which is a size-$n$ subset of $\{x_1, ..., x_m\}$. Denote this $X$ as $\{x_{j1}, ..., x_{jn}\}$. From the proof of inequality 2, we have that for any $\{\delta_1, ..., \delta_n\} \in \{\pm 1\}^n$, there exists $\theta \in (0, 1]$ such that for any $i$, we have that

$$|\sin(\frac{1}{\theta x_{ji}}) - \delta_i| \leq \frac{1}{2}.$$

On the other hand, the assumption $G(\mathcal{G}(X)) < \frac{n}{4}$ is equivalent to

$$G(\mathcal{G}(X)) = \mathbb{E}_\sigma \left[ \sup_{g \in \mathcal{G}} \sum_{i=1}^n \sigma_i g(x_{ji}) \right] < \frac{n}{4},$$

which implies that for $\epsilon_i$ to be iid Rademacher random variables

$$\mathbb{E}_\epsilon \left[ \sup_{g \in \mathcal{G}} \sum_{i=1}^n \epsilon_i g(x_{ji}) \right] < \frac{\sqrt{2\pi n}}{8} \leq \frac{3n}{8}$$

by the equivalence of Rademacher complexity and Gaussian average. In particular, there exists a set of $\{\epsilon_1, ..., \epsilon_n\}$ such that

$$\sup_{g \in \mathcal{G}} \sum_{i=1}^n \epsilon_i g(x_{ji}) < \frac{\sqrt{2\pi n}}{8} \leq \frac{3n}{8}.$$

Since $g(x) \in [-1, 1]$, a useful fact is that $\epsilon g(x) = 1 - |\epsilon - g(x)|$. To see this, when $\epsilon = 1$, the RHS is just $1 - (1 - g(x)) = g(x)$, the case of $\epsilon = -1$ is similar. As a result,

$$\sup_{g \in \mathcal{G}} \sum_{i=1}^n |\epsilon_i - g(x_{ji})| \geq \frac{5n}{8}.$$

From the discussion above we know that there exists $\theta_0$ such that for any $i$

$$|\sin(\frac{1}{\theta_0 x_{ji}}) - \epsilon_i| \leq \frac{1}{2}.$$

We set $\mu$ to be the induced distribution of sampling $x$ from $\{x_{j1}, ..., x_{jn}\}$ under a uniform distribution, with $y, z$ determined by $\theta^* = \theta_0$. We have the following estimation

$$\mathbb{E}_{(x,y,z)\sim\mu}\ell(g^*(x), z) = \frac{1}{n}\sum_{i=1}^n |\sin(\frac{1}{\theta_0 x_{ji}}) - g^*(x_{ji})|$$

$$\geq \frac{1}{n}\sup_{g \in \mathcal{G}}\sum_{i=1}^n |g(x_{ji}) - \epsilon_i| - \frac{1}{n}\sum_{i=1}^n |\sin(\frac{1}{\theta_0 x_{ji}}) - \epsilon_i|$$

$$\geq \frac{5}{8} - \frac{1}{2} = \frac{1}{8}.$$

Thus $H(\mu, \mathcal{G})$ is also lower bounded by $\frac{1}{8}$ in this case. Notice such lower bound is on the intrinsic gap $\mathbb{E}_{(x,y,z)\sim\mu}\ell(g^*(x), z) - \mathbb{E}_{(x,y,z)\sim\mu}\ell(f^*(x, y), z)$, it directly translates to a lower bound on the

actual risk as well. This concludes the first argument of the theorem, by setting $U$ as all possible distributions induced by a uniform distribution on a $n$-subset of $\{x_1, ..., x_m\}$ and any $\theta^* \in (0, 1]$.

Now we need to show that there exists a hypothesis $\mathcal{G}$ satisfying the second argument of the theorem. We simply choose $\mathcal{G} = \{g(x) = \theta x | \theta \in (0, 1]\}$. Clearly $\mathcal{R}(\mathcal{G}, S) = 0$ and with $g(x) = \theta^* x \in \mathcal{G}$ being the witness, which is exactly the output of the multimodal ERM algorithm on any sample.

$\square$

## D  Example of an $O(\sqrt{n})$ Advantage

Let $\mathcal{X} = [-1, 1] \subset \mathbb{R}$ and $\mathcal{Y} = \mathcal{B}_2 \subset \mathbb{R}^k$, we consider a distribution on $\mathcal{S}$, such that for any possible data point $(x, y, z) \in \mathcal{S}$, we have that $y = xv + y_0$ for some fixed $v, y_0 \in \mathcal{B}_1 \subset \mathbb{R}^k$. The hypothesis classes we consider are, all (projected) $k$-degree polynomials for $\mathcal{G}$

$$\mathcal{G} = \left\{ g(x) = \Pi_{\mathcal{B}_2}(\sum_{i=0}^{k} x^i v_i) | v_i \in \mathbb{R}^k \right\},$$

and all ($\epsilon$-smoothed) hyperplanes for $\mathcal{F}$

$$\mathcal{F} = \left\{ f(x) = \frac{x^\top v - c}{\max(|x^\top v - c|, \epsilon)} | v \in \mathcal{B}_1 \subset \mathbb{R}^{k+1}, c \in \mathbb{R} \right\},$$

where $\epsilon = o(1/\sqrt{k})$ is a constant. The projection and smoothness are not essential, they are only added to make the hypotheses consistent with the assumptions we made. We consider a sample size $n < k$. The worst-case Gaussian average $\max_{g \in \mathcal{G}} G(\mathcal{F}(S(g)))$ can be as large as $\Omega(n)$, because the set of all hyperplanes is known to be able to shatter any $k$ points in $\mathbb{R}^{k+1}$ that are linearly independent, while the class $\mathcal{G}$ is expressive enough to contribute a worst-case $g$ that does map $X$ to a set of linearly independent points. In particular, let's consider a uniform distribution on $\mathcal{X}$. With probability 1 there is no duplicate in $X$, and there exists $g \in \mathcal{G}$ such that $g(x_i) = e_i$ for all $i = 1, ..., n$ because the linear system has more variables than constraints. Then for each possible partition of $\{e_i | i = 1, ..., n\}$, there is a hyperplane with $\Theta(\frac{1}{\sqrt{n}})$ margin separating the two parties. As a result, since $\mathcal{F}$ contains hyperplanes with margin $2\epsilon = o(1/\sqrt{n})$, it shatters $X$ and $G(\mathcal{F}(S(g))) = \Omega(n)$.

However, it's possible that the learnt $\hat{g}$ is a low-dimensional polynomial. In particular, when it indeed learns the correct connection $g(x) = xv + y_0$, then the $n$ points $(x_1, \hat{g}(x_1)), ..., (x_n, \hat{g}(x_n))$ are on the same line, and $G(\mathcal{F}(\hat{X}, \hat{Y}))$ is just $O(\sqrt{n})$. Note that learning such $\hat{g}$ via the multimodal ERM algorithm only requires $m = \Omega(k^2)$ in general since $\sum_{i=0}^{k} x^i v_i$ represents a $k \times (k+1)$ linear system.

The only potential question left is the Lipschitz constant $L$ of $\mathcal{F}$, here being $1/\epsilon$. However, although $L$ appears in the upper bound of Theorem 4, it's not multiplied to the leading term $\sum_{t=1}^{T} G(\mathcal{F}(\hat{X}_t, \hat{Y}_t))$, instead it's mitigated by the large $m$. Hence the comprasion between $G(\mathcal{F}(\hat{X}, \hat{Y}))$ and $\max_{g \in \mathcal{G}} G(\mathcal{F}(S(g)))$ is fair regardless of the large Lipschitz constant $L$.

We note that this example doesn't contradict the results of [22, 38]. It's known that in general $G(\mathcal{F}(X)) = \tilde{O}(\sqrt{nd})$ where $d$ is the VC-dimension of $\mathcal{F}$. In this example, $d \geq k \geq n$, thus the worst-case value $\max_{g \in \mathcal{G}} G(\mathcal{F}(S(g)))$ actually matches the upper bound $\tilde{O}(\sqrt{nd})$. However, the term $G(\mathcal{F}(\hat{X}, \hat{Y}))$ in our bound can be smaller by a factor of $O(\sqrt{n})$.

## E  Example of Separable Data

Consider $\mathcal{X} = \mathcal{Y} = [0, 1]$, equipped with a strictly-increasing function $f : [0, 1] \to [0, 1]$ satisfying $f(0) = 0, f(1) = 1$. Any potential observation from $\mathcal{S}$ is governed by the following:

$$y = f(x), z = \mathbf{sign}(x - y).$$

Clearly the function $f$ serves as a bijective connection mapping between $\mathcal{X}, \mathcal{Y}$. We notice that any sample $S$ on both modalities $\mathcal{X}, \mathcal{Y}$ with any districution $\mu$ is linearly separable, by a linear function $x - y = 0$.

In contrast, the decision region of either $\mathcal{X}$ or $\mathcal{Y}$ can be arbitrary finite union of open intervals in $[0, 1]$, by the definition of $f$. In particular, for any target decision boundary as a finite subset $A = \{a_1, ..., a_n\}$ of $[0, 1]$, we can construct $f$ such that

$$\{x | f(x) = x\} = A.$$