# OpenReview forum: "A Theory of Multimodal Learning"
_NeurIPS.cc/2023/Conference — NeurIPS 2023 poster_

### Official Review · Reviewer_XfDh · 2023-06-24

**Soundness:** 3 good
**Presentation:** 2 fair
**Contribution:** 2 fair
**Rating:** 5
**Confidence:** 2

**Summary:**

This paper aims to provide a theoretical framework for multimodal learning, that aims to explain why multimodal learning is sometimes better than unimodal learning even when the model is only applied on unimodal tasks. To do this, the authors measure the generalization bounds of multimodal learning on the excess population with Gaussian average, and showed multimodal has a superior generalization bound. They also try to formulate the problem with connection and heterogeneity, and show when multimodal is better than unimodal under their evaluation metrics.

**Strengths:**

1. Motivation is strong. Theoretical understanding of multimodal learning is an important yet less investigated problem.
2. The theoretical investigation is interesting and relatively novel to me. It is particularly interesting to characterize/formulate the multimodal problem into connection and heterogeneity.

**Weaknesses:**

1. The authors nicely concluded their weakness in their limitations section, I think the below points are the major weakness that are non-neglectable (although pointed out in the final section): The work needs more natural assumptions, it currently requires too many hypothesis, and it needs more realistic examples.
2. Related to the above point -- I think the results would be much stronger if it can be backed up by some toy experiments with synthetic datasets to demonstrate how the proof are useful in real-world cases. The current theoretical results are nice, but that content is not sufficient enough to be published on its own for NeurIPS.
3. The presentation/writing of the paper could be further improved (by a lot). e.g. Line 17 ’Diamond Sutra’; line 224 "The story is not yet complete"; line 245 "Let’s".

Overall, I think this is an interesting work that needs further iterations/improvements to be able to be published at NeurIPS.

**Questions:**

NA

**Limitations:**

The authors adequately addressed the limitations.

---

> ### Author Rebuttal · Authors · 2023-08-03
>
> We appreciate the constructive feedback and the thoughtful questions posed. We'll address each of your concerns below.
>
> **More natural assumptions:** our work is devoted to providing a general theory of multi-modal learning, which inevitably comes at the cost of some loss in practicality. We feel the Lipschitz assumption is the major one, which other reviewers also agree upon. In general, we believe it's necessary to jump out of the current hypothesis-composition mindset to relax the Lipschitz assumption, since this assumption is also known to be a fundamental limit of representation learning generalization bounds.
>
> Within our existing framework, assuming zero training loss of $\mathcal{G}$ on $X'$ ($\mathcal{R}(\mathcal{G},X')=0$ which is natural in practice) can partially mitigate this problem. The $L$-dependent term is then $O(\frac{L}{\sqrt{m}})$ where $m$ the size of unlabeled data is huge enough to lighten the effect of $L$ (see line 444). Moreover, if we consider the supervised setting, assuming zero training loss will immediately reduce the $L$-dependent term to zero. Future explorations could focus on modifying the analysis for this setting.
>
> **Toy experiments:** in our opinion, the benefit of toy experiments with synthetic data somewhat marginal. Our theory is on explaining an existing phenomenon instead of proposing a better algorithm, thus re-validating the theory seems less significant to us: toy experiments can only serve as a sanity check to what our theory has rigorously proven. On the other hand, large-scale experiment is beyond the scope of the current work, and such phenomenon is already widely observed on large-scale practices of multi-modal learning. For these reasons, we believe experiment is more of a complement rather than a necessity to our theory.
>
> Is there a specific experimental setup that the reviewer would like to see?
>
> **Theory is not sufficient:** we believe the theoretical results are significant enough on their own. Here are a few points raised also by you and other reviewers: the problem studied is important yet less investigated, which became even more urgent due to recent empirical success; the proposed theory is intuitive and original; the technical contributions are solid.
>
> **Writing:** thank you for catching some typos we missed. We will correct them and make a pass of the paper to improve writing.

---

> > ### Comment · Reviewer_XfDh · 2023-08-18
> > **Response to rebuttal**
> >
> > Thank you for the detailed response. I read the other reviewers' comments as well and agreed that the theory, as it is rigorously proven, might be sufficient by itself. The experimental setup that I would like to see as a reader and as a reviewer is also only toy experiments, which might be compensated by the simple examples at the beginning as pointed out by other reviewer.
> >
> > However, as I personally am not familiar with the other works assumptions well enough, I will leave it for the AC to decide (1) if the assumptions used in the paper are natural/general enough such that the theoretical results are of sufficient value; (2) As the paper considers multimodal generalization properties instead of multimodal optimization, if the corresponding results are of sufficient value.
> >
> > Thus I will increase my score from 4 to 5 and decrease my confidence score from 3 to 2.

---

> > > ### Author Response · Authors · 2023-08-18
> > >
> > > Thanks for your feedback! We will add an experiment on the example used in Section 2.1 as you suggested (it's very simple and will be easy to implement).

---

### Official Review · Reviewer_wXhy · 2023-06-24

**Soundness:** 3 good
**Presentation:** 3 good
**Contribution:** 3 good
**Rating:** 7
**Confidence:** 3

**Summary:**

This paper introduces a theoretical framework aimed at elucidating the phenomenon wherein a multimodal neural network, trained on multimodal data, can exhibit strong performance on unimodal data. The framework incorporates Gaussian averaging and operates within a semi-supervised context. The findings demonstrate that under the conditions of heterogeneity and connectivity, it is possible to identify a bridging function $\hat{g}$ and a projection function $\hat{f}$ that enable comparable generalization error on unimodal data, as if the network were provided with multimodal data.

**Strengths:**

Clear presentation, nice illustrative sinusoidal function, solid technical contribution.

**Weaknesses:**

Some clarifications are needed.

**Questions:**

Overall, I believe the paper is of good quality and deserves acceptance. The questions I'm about to raise are intended for discussion purposes and should be considered by the authors. Lastly, I have some suggestions for improving the writing, particularly in areas that could benefit from greater clarity, especially for readers encountering the content for the first time.

1) I recommend that the authors rephrase the explanations regarding Equation (1). Based on Section 3 with T=1, my understanding is that we solve for $\hat{f_1}$ and $\hat{g}$ using ERM, substituting them back into $L(\hat{g},\hat{f}_1)$ as shown between lines 192 and 193, which is identical to Equation (1). In other words, we learn $\hat{f}_1$ and $\hat{g}$ on multimodal data. However, the current wording in line 137 states that we are attempting to minimize Equation (1) with respect to $\hat{f}$ and $\hat{g}$, which may confuse readers. When I first encountered Equation (1), I assumed that the second term had no impact because it does not depend on $\hat{f}$ and $\hat{g}$, leading me to believe that Equation (1) simplifies to solving for $\hat{f}$ and $\hat{g}$ based solely on $x$.

2) If my understanding is correct, I suggest explicitly writing out $s_{t1}=(x_{t1},y_{t1},z_{t1})$ in line 186.

3) The learning framework presented in this section assumes learning the bridging function $\hat{g}$ on a set of unlabeled data and the projection functions $\hat{f_t}$ on the labeled dataset. While this is certainly a valid learning framework, it seems to me that the most common one might be a fully supervised setting where paired data $(x_i, y_i, z_i)$ are given.

4) In line 8, the authors state that a multimodal network trained on a multimodal dataset can outperform a unimodal network even on unimodal tasks, and this paper aims to demonstrate that. However, in my understanding, this paper actually shows that a multimodal network can perform comparably on a unimodal dataset **as if** multimodal data is provided, as indicated by Equation (1). These two statements differ.

5) The model considered by the authors takes the form of $f(x,y\approx g(x))$. It seems to deviate from what is typically used in the literature. A common form found in the literature is $f(g(x),h(y))$, where $g$ and $h$ are two encoders, and $f$ could represent early/middle/late fusion.

6) I would like to bring an important topic to the attention of the authors. There have been several explorations on multimodal knowledge distillation. Under this topic, it would be highly interesting to theoretically analyze the performance of multimodal (or unimodal) teachers (or students). For further reference, please see the following paper: https://arxiv.org/abs/2206.06487. It would be fascinating to extend this work to knowledge distillation settings as a future area of research.

7) I would suggest the authors to include literature on multi-view learning as Ref [16] did. Because that is closely related to multimodal learning, or some might argue their theory are essentially the same.

8) It will be good to include numerical experiments to support the theorems (or at least the claim in line 258-260).

**Limitations:**

The authors clearly stated the limitation of the work in the last section.

---

> ### Author Rebuttal · Authors · 2023-08-03
>
> We greatly appreciate your thorough feedback and constructive suggestions! We will address each of your questions individually and please feel free to let us know if you have further concerns.
>
> **Q1:** thank you for highlighting this. Your understanding is accurate, and we will revise our language to make this clearer and prevent potential confusion.
>
> **Q2:** we agree with your suggestion and will incorporate this change in our revised manuscript.
>
> **Q3:** we agree that a fully supervised setting is more common, however, we decided to explore the more general semi-supervised setting for a few reasons: first, using unlabeled data can provide sharper bounds when available and utilized; second, in practice, labeled multi-modal data is rarer than unlabeled data, and the latter is crucial to the empirical success of recent large-scale multi-modal models; lastly, the supervised setting can be subsumed by the semi-supervised setting by partitioning the data.
>
> **Q4:** you have precisely interpreted what Theorem 4 states. However, in line 8, we were also implicitly taking the separation between multi-modal and uni-modal (Theorem 7) into account. We will make the statement more precise.
>
> **Q5:** we focus on a general theory, thus we only consider the simplest case $f(x,y)$. Notice that the encoder form can be subsumed by the general form, by setting the hypothesis class to include $f(g,h)$. This opens up an interesting research question, that under what assumption (for example, when data is low-rank) the encoder form can lead to better generalization bounds.
>
> **Q6:** we greatly appreciate your suggestion. We will delve into the referenced materials and investigate the potential application of our theory in knowledge distillation.
>
> **Q7:** we will expand our discussion on multi-view learning, and make clear distinctions where necessary as you suggested.
>
> **Q8:** we feel re-validating the theorems with toy experiments somewhat marginal, as this work provides a theoretical explanation   rather than a better algorithm. On the other hand, large-scale experiment is beyond the scope of the current work. We agree experiments might be more useful for the principle part (line 258-260), but for practical purposes modification of the algorithm is needed, and we leave it to future study.

---

> > ### Comment · Reviewer_wXhy · 2023-08-12
> > **Thanks for the clarification**
> >
> > I value the clarifications provided by the authors and find no more questions. To recap, this paper adds valuable contributions to the theoretical realm of multimodal learning. Considering potential subjective preferences, I am inclined to believe that the mathematical rigor alone warrants its publication.

---

> > > ### Author Response · Authors · 2023-08-12
> > >
> > > Thanks for your evaluation and support of our work! All the clarifications will be reflected in this paper accordingly.

---

### Official Review · Reviewer_TbhH · 2023-07-09

**Soundness:** 3 good
**Presentation:** 4 excellent
**Contribution:** 3 good
**Rating:** 8
**Confidence:** 2

**Summary:**

This study proposes a new theoretical foundation for multimodal learning. In particular, regarding the phenomenon that models trained in multiple modalities perform better on single-modality tasks than fine-tuned single-modality models, this paper proposes that multimodal learning is a composition of connection and label predictor, and shows that generalization performance is better than single-modality learning when both connection and heterogeneity are present.

**Strengths:**

- This paper is well-organized and very easy to understand.
- Although I am not an expert in learning theory, I have checked throughout and found the theory to be valid. In particular, it is interesting and novel to compare the generalization bound with the usual unimodal learning case by setting multimodal learning as a composition of connection and label predictor.

**Weaknesses:**

- In this study, multimodal learning is considered to be prediction from unimodal data, consisting of prediction by connections from one modality to another and label prediction from them. However, this setting may appear somewhat unnatural when viewed from the perspective of normal multimodal learning. It would be valuable to have a more in-depth discussion on this aspect.
- In multimodal learning, it is common to consider more than two modalities. It would be even more insightful to explore the potential implications and extensions of this theory when applied to settings involving multiple modalities.

**Questions:**

I would appreciate it if the authors could respond to the above-mentioned weaknesses.

**Limitations:**

The authors explain the limitations of this theory very clearly.

---

> ### Author Rebuttal · Authors · 2023-08-03
>
> We appreciate your constructive feedback and inquiries!
>
> **Unnatural setting:** we agree that the mapping + predictor learning process we described, may not perfectly align with practical multi-modal learning scenarios. However, the goal of our work is to propose a general theoretical framework, therefore clarity comes with a degree of departure from real-world applicability. The perspective we offer on multi-modal learning is one of many, and we believe that it provides an intuitive way to understand the mechanisms underlying the success of multi-modal learning.
>
> In addition, beyond the theorem statement, our theory may have implications to more general scenarios: the process of learning connections between modalities could be implicitly occurring in empirical multi-modal learning. This poses an intriguing research question: can we establish similar generalization bounds for a more practical "one-pass" algorithm, which does not require learning connections and predictors separately?
>
> **Multiple modalities:** this work is focused on the case of two modalities for the sake of clarity, but our theoretical framework can be extended to accommodate multiple modalities in a similar way. In this case, the ERM algorithm would learn a mapping from a subset of modalities to all modalities, which involves only one hierarchy as in the two-modality case. Therefore our analysis naturally carries over to this new setting. In particular, the hypothesis class $\mathcal{G}$ will include mappings from the subset to all modalities.
>
> If a more complicated relationship graph between multiple modalities with more hierarchies is considered and the data don't contain full modalities, we believe new tools beyond the current representation learning framework are required, potentially from the field of graph theory.

---

### Official Review · Reviewer_iaEv · 2023-07-10

**Soundness:** 3 good
**Presentation:** 2 fair
**Contribution:** 3 good
**Rating:** 6
**Confidence:** 4

**Summary:**

The paper establishes theoretical bounds for generalization in multimodal learning, where functions mapping between two modalities and to the label are learned. The authors demonstrate that multimodal learning achieves better generalization by decoupling the learning of hypotheses and provide insights regarding the connection and heterogeneity between modalities.

**Strengths:**

1.	The paper aims at addressing an important topic and offers theoretical insights into the superiority of multimodal learning over unimodal learning, particularly regarding the decoupling of hypothesis learning.
2.	The use of simple examples at the beginning enhances the readability of the paper and helps illustrate certain aspects of the main ideas.

**Weaknesses:**

1.	The motivating example appears to be restrictive as it relies on F containing only the ground truth mapping from modality Y to label Z, and by the construction the map from Y to Z is exatly the only `hard part' of the learning. It might be slightly better to use the harder example in Remark 2. How would the comparison between multimodal and unimodal learning be in this case?
2.	Theorem 7 only shows the existence of cases where multimodal learning surpasses unimodal learning, without delving into a deeper discussion on the required conditions for such cases. As a result, it remains unclear whether these instances solely rely on trivial constructions, such as letting F be a class containing a single function, which is exactly the true map between Y and Z, with learning class G (map between X and Y) constructed as a trivial task. More importantly, from proof of theorem 7, this is exactly how the existence is proved, i.e., by taking the motivating example in section 2.1. Such constructions imply that (1) learning the mapping between modalities is exceedingly simple, and (2) one can directly determine the relationship between a modality and the labels without the need for learning, both of which are impossible in practice. This greatly limits the paper's ability to capture the underlying mechanisms behind the success of multimodal learning. On direction for improvement could be to seek for more realistic instances.
3.	The authors claim that heterogeneity (e.g., lines 167, 255) is one of the two factors leading to the superiority of multimodal learning. However, heterogeneity appears to be more of a consequence rather than the underlying reason. The statement in line 167, explaining heterogeneity as 'multimodal data being easier to learn than unimodal'. Furthermore, in definition 6, heterogeneity is directly defined as the difference between population risks of learning with single and multimodal approaches. This raises my concerns about circular reasoning within the argument. As a result, section 4 and the arguments regarding heterogeneity currently lack clarity and informativeness.

**Questions:**

In addition to the questions raised in Weaknesses, I am also curious if the framework presented in the paper can offer insights or potentially be extended to provide analysis for other methods, such as CLIP, where the objective is to learn shared representations for both modalities instead of learning a map from one modality to another.

**Limitations:**

The authors have adequately discussed limitations in the paper.

---

> ### Author Rebuttal · Authors · 2023-08-03
>
> We appreciate the constructive feedback and the thoughtful questions posed. We'll address each of your concerns below.
>
> **The choice of the example:** we discussed the simpler example purely for the sake of clarity, and the harder example in Remark 2 is strictly stronger because the separations between Gaussian averages hold for **both** modalities simultaneously. A slight modification to the construction of Remark 2 can fully recover such separations for both modalities as in the simpler example. Details are given below.
>
> Any potential data point $(x,y,z)$ is now generated by three parameters $c\in (0,1), \theta_1 \in (1,2),\theta_2 \in (-2,-1)$, under the constraint that $\theta_1+\theta_2\ne 0$, and $(x,y,z)$ is of form $(c\theta_1,c\theta_2,c(\theta_1+\theta_2))$. For the learning problem, parameters $\theta_1,\theta_2$ are pre-fixed and unknown, data is then generated from some distribution of $c$. The hypothesis classes are now $\mathcal{G}=\{g(x)=\theta x, \theta \in (-1,0)\cup(0,1)\}$ and $\mathcal{F}=\sin (1/x)$.
>
> For any uni-modal data $x=c\theta_1$, the range of ratio $(x+y)/x$ is $(1-2/\theta_1,0)\cup(0,1-1/\theta_1)$. This range is a subset of $(-1,0)\cup(0,1)$ and we have that $\max(|1-2/\theta_1|,|1-1/\theta_1|)\ge (2/\theta_1-1+1-1/\theta_1)/2\ge 1/4$. As a result, $G(\mathcal{F} \circ \mathcal{G}(X))$ in this case is at least $1/4$ of that in the simpler example because the flip of sign doesn't affect Gaussian averages, thus the term remains $\Omega(n)$. On the other hand, we have that $\max(|1-2/\theta_1|,|1-1/\theta_1|)\le 1$, so $G(\mathcal{G}(X))=O(\sqrt{n})$ holds still. The same argument holds for $\mathcal{Y}$ as well since it mirrors $\mathcal{X}$.
>
> **Lower bound conditions:** in our understanding, your question is can we go beyond worst case constructions and derive a general condition for such separation? While our current construction seems simple, it effectively serves its purpose in illustrating a lower bound. Deriving general instance-dependent separation bounds seems challenging, since this task is not much easier than deriving closed-form estimations of the generalization error. Nevertheless, we agree that more realistic examples can be more insightful. A potential method is to trade some flexibility from $\mathcal{G}$ to $\mathcal{F}$. We can also modify the example in Appendix D which is used for a similar purpose, but this example is less intuitive than the one in the main-text and it has other restrictions.
>
> **The role of heterogeneity:** we acknowledge your concern about the role of heterogeneity, and whether it forms circular reasoning. Let's briefly clarify the logic behind. The superiority of multi-modal model over uni-modal model on uni-modal tasks consists of two parts: multi-modal model is comparable to uni-modal model as if multi-modal data is provided (connection), and multi-modal data is easier to learn than uni-modal by **any** uni-modal ERM algorithm (heterogeneity). Therefore, the superiority is the consequence of the two factors connection and heterogeneity.
>
> Why heterogeneity isn't the consequence of such superiority? Because heterogeneity has stronger requirements: from Definition 6, heterogeneity is defined to be the gap for **any** uni-modal ERM algorithm, while for a particular algorithm with such superiority, the hypothesis class in use doesn't exclude the possibility that there exists better hypothesis classes with a smaller gap. To sum up, connection + heterogeneity is a sufficient condition for superiority, while superiority alone doesn't necessarily imply heterogeneity, and Theorem 7 shows the existence of a scenario with connection + heterogeneity. We will clarify and emphasize this point to avoid potential confusion.
>
> **Extension to CLIP:** we appreciate your suggestion to explore our theory's applicability to CLIP. There are similarities between our work and the approach CLIP employs, although a more nuanced analysis is needed due to CLIP's use of cosine similarity as a specific measure. We'll delve deeper into this in our future work.

---

> > ### Comment · Reviewer_iaEv · 2023-08-12
> > **Thanks for your response**
> >
> > Thanks for your response!  I'd like to better grasp the logic behind the role of heterogeneity. From my understanding so far, the notion of the "heterogeneity gap" assumes that the "population risk of learning a single modality is notably higher than that of learning both modalities, for any algorithm". In your argument, it seems you wanted to establish that this would lead to the conclusion that "population risk of learning a single modality is notably higher than that of learning both modalities, for a particular algorithm". I'm curious why the transition from the heterogeneity gap to this superiority isn't trivial, as the conclusion appears to be inherently contained within the definition of the heterogeneity gap.
> >
> > I'm also interested in understanding how the joint impact of heterogeneity and connection contributes to this superiority. In Theorem 4, it appears that heterogeneity alone is sufficient.

---

> > > ### Author Response · Authors · 2023-08-12
> > >
> > > Thanks for your questions. Let's start with a brief recap on the problem setting and definitions.
> > >
> > > **The learning task:** given uni-modal population data from $\mathcal{X}$, predict the true label $z$.
> > >
> > > **The goal of this work:** to show for the above task, a model trained on multi-modal data $(X,Y)$ outperforms any model trained on single-modal data $X$, on the same uni-modal population data from $\mathcal{X}$.
> > >
> > > **Heterogeneity:** there is a model with multi-modal data from $(\mathcal{X,Y})$, which outperforms any model with uni-modal data from $\mathcal{X}$.
> > >
> > > **Connection:** a mapping between modalities $\mathcal{X,Y}$ is learnable.
> > >
> > > For your first question, in the definition of heterogeneity, the multi-modal algorithm is given multi-modal data $(X,Y)$ during both training and testing, while the task only provides the algorithm with uni-modal data in the testing phase. Therefore, heterogeneity alone can't fulfill the goal because it has a stronger requirement.
> > >
> > > For your second question, a good connection implies that for an algorithm trained on multi-modal data $(X,Y)$, when faced with uni-modal population data from $\mathcal{X}$, it has comparable performance to as if multi-modal population data from $(\mathcal{X,Y})$ is given. Combined with heterogeneity, it fulfills the goal.

---

> > > > ### Comment · Reviewer_iaEv · 2023-08-14
> > > > **Thanks for your response**
> > > >
> > > > Thank you for the clarification. I now have a better understanding of the entire narrative. However, I'm still curious about how **the combined impact** of connection and heterogeneity is demonstrated in the theorems. To my understanding, Theorems 4 and 7 each illustrate that connection/heterogeneity individually results in a low excess risk, but please correct me if I'm wrong. Could the author provide further explanation? For instance, in Theorem 7, where does connection come into play and contribute to achieving $L(\hat{g}, \hat{f})=0$? Additionally, in the case of Theorem 4, it seems that good connection alone might already let the excess risk vanish. How do we ascertain that we also require heterogeneity in this context?

---

> > > > > ### Author Response · Authors · 2023-08-14
> > > > >
> > > > > Thanks for your further questions! We will clarify each of them, detailed below.
> > > > >
> > > > > Recall our goal is to show a model trained on multi-modal data outperforms any model trained on single-modal data, on an uni-modal learning task. This goal can be decomposed into two parts (abbreviation here for clarity): model w. MM train + UM test $\approx$ model w. MM train + MM test, and model w. MM train + MM test > model w. UM train + UM test. The first part corresponds to connection while the second part corresponds to heterogeneity.
> > > > >
> > > > > **Theorems 4 and 7 imply low excess risk individually:** your understanding about Theorem 4 is correct, which illustrates that connection itself can result in a low excess risk. However, Theorem 7 is about connection + heterogeneity, instead of heterogeneity alone.
> > > > >
> > > > > Theorem 7 doesn't aim to show a general condition under which a low excess risk is achievable. By the definition of heterogeneity, as long as the heterogeneity gap exists, it directly implies that model w. MM train + UM test > model w. UM train + UM test as long as Theorems 4 admits a low risk. Recall the risk in Theorems 4 vanishes if and only if $\mathcal{R}(\mathcal{G},S')=0$ which means a perfect connection. Thus, the only question left is whether such desired scenario (large heterogeneity gap + perfect connection) actually exists.
> > > > >
> > > > > To this end, Theorem 7 provides an instance, in which not only the connection is perfectly learnable (the second sentence), but the heterogeneity gap is large (the first sentence), proving that our theory is indeed effective.
> > > > >
> > > > > **The role of connection in Theorem 7:** connection is defined as the approximate realizability $\mathcal{R}(\mathcal{G},S')$. In the instance of Theorem 7, this term equals zero, thereby provides an $o(1)$ upper bound on $L(\hat{g},\hat{f})$. Because the instance we consider is simple, we are able to further achieve the stronger goal of deriving the actual value of $L(\hat{g},\hat{f})$, which is zero.
> > > > >
> > > > > **Connection implies vanishing risk alone:** in Theorem 4, the excess risk vanishes w.r.t. $n,m,T$, if and only if $\mathcal{R}(\mathcal{G},S')=0$. Thus a good connection alone does imply a vanishing risk. However, such risk is defined to compare with model w. MM train + MM test, while our ultimate goal is to compare with model w. UM train + UM test and illustrate a gap between risks.
> > > > >
> > > > > **Illustrating examples:** we refer the reviewer to the two simple examples (line 169-174) for a very intuitive explanation why both connection and heterogeneity are necessary.
> > > > >
> > > > > **Summary:** Theorem 4 provides a general risk bound w.r.t. connection, which is meaningful on its own. Theorem 7 combines heterogeneity and Theorem 4 to show our theory provides an effective explanation to the practical phenomenon of interest.

---

> > > > > > ### Comment · Reviewer_iaEv · 2023-08-20
> > > > > >
> > > > > > Thank you for your detailed clarification. The logical flow you've provided, particularly how the theorems align with the objectives, may not be immediately evident and requires careful yet intuitive elaboration in the paper for readers to fully grasp. It would be helpful to add the above clarification in the revised paper. That being said, following your explanation, I now see the contribution of the paper and have decided to raise the score accordingly.

---

> > > > > > > ### Author Response · Authors · 2023-08-20
> > > > > > >
> > > > > > > We will add the above explanations in the new version of this paper as you suggested. Thanks again for the discussion!

---

### Official Review · Reviewer_PsHQ · 2023-07-12

**Soundness:** 4 excellent
**Presentation:** 3 good
**Contribution:** 4 excellent
**Rating:** 8
**Confidence:** 3

**Summary:**

The authors presents an interesting theoretical framework that allow them to estimate a generalisation bound for multimodal learning.
The main result consist in proving the the bound for the multimodal case is superior to the unimodal one up to a factor $O(\sqrt{n})$ that depends on the sample size of the dataset considered.

**Strengths:**

The paper is pleasant to read and the work is refreshingly original. Moreover, even if at an abstract level, it builds a theoretical framework for multimodal learning that was long missing in the community, especially given the importance and the relevance that multimodal models  are acquiring in recent times.

**Weaknesses:**

As also the author points out the results presented are rather abstract and more fine-grained analysis of specific multi-modal learning algorithms would be of broad interest.
Another note is the notation usage, some key terms and notations should be briefly explained (it is sufficient in the appendix) to allow less experienced readers to follow the paper more straightforwardly.

**Questions:**

Some questions that I would like to have feedback from the others on are:
- How would you proceed to relax the condition of having a predictor class containing not only Lipschitz functions?
- How would you generalise the results presented to the case of considering more than two modalities? Namely having $m$ modalities with multiple mapping functions linking single and combination of modalities. Do you see any specific limitation in the current framework?

**Limitations:**

First of all, the authors did an amazing job in highlighting the limitations of their work in their manuscript.
Here I simply highlight the major ones:
- Lipschitz assumption on $\mathcal{F}$ (predictor class) are restrictive as it prevents a direct extension to DNN
- Lack of a concrete analysis for specific multimodal learning algorithms

---

> ### Author Rebuttal · Authors · 2023-08-03
>
> We appreciate your thoughtful feedback and constructive suggestions!
>
> **Fine-grained analysis:** the scope of this current study is primarily to establish a general theoretical framework for multi-modal learning. We will certainly delve into specific algorithms based on our framework in future works.
>
> **Notation usage:** we will clarify the key terms and notations to ensure they are clearly explained.
>
> **Lipschitz assumption:** in general, we believe it's necessary to jump out of the current hypothesis-composition mindset to relax the Lipschitz assumption, since this assumption is also known to be a fundamental limit of representation learning generalization bounds.
>
> Within our existing framework, assuming zero training loss of $\mathcal{G}$ on $X'$ ($\mathcal{R}(\mathcal{G},X')=0$ which is natural in practice) can partially mitigate this problem. The $L$-dependent term is then $O(\frac{L}{\sqrt{m}})$ where $m$ the size of unlabeled data is huge enough to lighten the effect of $L$ (see line 444). Moreover, if we consider the supervised setting, assuming zero training loss will immediately reduce the $L$-dependent term to zero. Future explorations could focus on modifying the analysis for this setting.
>
> **Multiple modalities:** this work is focused on the case of two modalities for the sake of clarity, but our theoretical framework can be extended to accommodate multiple modalities in a similar way. In this case, the ERM algorithm would learn a mapping from a subset of modalities to all modalities, which involves only one hierarchy as in the two-modality case. Therefore our analysis naturally carries over to this new setting. In particular, the hypothesis class $\mathcal{G}$ will include mappings from the subset to all modalities.
>
> If a more complicated relationship graph between multiple modalities with more hierarchies is considered and the data don't contain full modalities, we believe new tools beyond the current representation learning framework are required, potentially from the field of graph theory.

---

> > ### Comment · Reviewer_PsHQ · 2023-08-12
> > **Thanks for the clarifications and the response**
> >
> > The authors properly addressed my questions and I confirm my evaluation of a strong accept.

---

> > > ### Author Response · Authors · 2023-08-12
> > >
> > > Thanks for your evaluation and support of our work!

---

### Decision · Program_Chairs · 2023-09-21

**Decision:**

Accept (poster)

**Comment:**

The paper provides a theoretical framework for multimodal learning and shows that when functions mapping between two modalities and to the label are learned, multimodal learning achieves a genearlization performance that is superior to the unimodal one up to a factor that depends on the square root of the sample size. I believe the paper provides valuable insights for the community and can be accepted as is. However, I strongly encourage the authors to incorporate the clarifications on the theory suggested by iaEv, discussion on and comparison to multi-view learning suggested by wXhy, and add an experiment on the example used in Section 2.1 suggested by XfDh. These will greatly improve the clarify of the paper.